# Anti-Tumor Activity of Orally Administered Gefitinib-Loaded Nanosized Cubosomes against Colon Cancer

**DOI:** 10.3390/pharmaceutics15020680

**Published:** 2023-02-17

**Authors:** Ahmed A. El-Shenawy, Mahmoud M. A. Elsayed, Gamal M. K. Atwa, Mohammed A. S. Abourehab, Mohamed S. Mohamed, Mohammed M. Ghoneim, Reda A. Mahmoud, Shereen A. Sabry, Walid Anwar, Mohamed El-Sherbiny, Yasser A. Hassan, Amany Belal, Abd El hakim Ramadan

**Affiliations:** 1Department of Pharmaceutics and Pharmaceutical Technology, Faculty of Pharmacy, Al-Azhar University, Assiut Branch, Assiut 71524, Egypt; 2Department of Pharmaceutics and Clinical Pharmacy, Faculty of Pharmacy, Sohag University, Sohag 82524, Egypt; 3Department of Biochemistry, Faculty of Pharmacy, Port Said University, Port Said 42515, Egypt; 4Department of Pharmaceutics, Faculty of Pharmacy, Umm Al-Qura University, Makkah 21955, Saudi Arabia; 5Department of Pharmacy Practice, College of Pharmacy, AlMaarefa University, Ad Diriyah 13713, Saudi Arabia; 6Pharmacognosy and Medicinal Plants Department, Faculty of Pharmacy, Al-Azhar University, Cairo 11884, Egypt; 7Department of Pharmaceutics, Faculty of Pharmacy, Zagazig University, Zagazig 44519, Egypt; 8Department of Pharmaceutics and Pharmaceutical Technology, Faculty of Pharmacy, Al-Azhar University, Cairo 11751, Egypt; 9Department of Basic Medical Sciences, College of Medicine, AlMaarefa University, P.O. Box 71666, Riyadh 11597, Saudi Arabia; 10Department of Anatomy, Faculty of Medicine, Mansoura University, Mansoura 35516, Egypt; 11Department of Pharmaceutics, Faculty of Pharmacy, Delta University for Science and Technology, Gamasa 35712, Egypt; 12Medicinal Chemistry Department, Faculty of Pharmacy, Beni-Suef University, Beni-Suef 62514, Egypt; 13Department of Pharmaceutical Chemistry, College of Pharmacy, Taif University, P.O. Box 11099, Taif 21944, Saudi Arabia; 14Department of Pharmaceutics, Faculty of Pharmacy, Port Said University, Port Said 42515, Egypt

**Keywords:** gefitinib, nanosized cubosomes, nanoparticles, colon cancer, gene expression, histopathological examination

## Abstract

Gefitinib (GFT) is a tyrosine kinase inhibitor drug used as a first-line treatment for patients with advanced or metastatic non-small cell lung, colon, and breast cancer. GFT exhibits low solubility and hence low oral bioavailability, which restricts its clinical application. One of the most important trends in overcoming such problems is the use of a vesicular system. Cubosomes are considered one of the most important vesicular systems used to improve solubility and oral bioavailability. In this study, GFT cubosomal nanoparticles (GFT-CNPs) were prepared by the emulsification method. The selected formulation variables were analyzed and optimized by full factorial design and response surface methodology. Drug entrapment efficiency (EE%), transmission electron microscopy, particle size, polydispersity index, in vitro release and its kinetics, and the effect of storage studies were estimated. The chosen GFT-CNPs were subjected to further investigations as gene expression levels of tissue inhibitors of metalloproteinases-1 (TIMP-1) and matrix metalloproteinases-7 (MMP-7), colon biomarkers, and histopathological examination of colon tissues. The prepared GFT-CNPs were semi-cubic in shape, with high EE%, smaller vesicle size, and higher zeta potential values. The in vivo data showed a significant decrease in the serum level of embryonic antigen (CEA), carbohydrate antigen 19-9 (CA 19-9), and gene expression level of TIMP-1 and MMP-7. Histopathological examination showed enhancement in cancer tissue and highly decreased focal infiltration in the lamina propria after treatment with GFT-CNPs.

## 1. Introduction

Cancer develops once cells in one region of the body proliferate rapidly in an uncontrolled manner [1]. The malignant cells can infiltrate nearby healthy tissue and kill it [2,3]. Colorectal cancer (CRC) is the third most common cancer after lung and breast cancer and the third most common cause of death due to cancer globally [4,5]. Each year, approximately 1 million new CRC cases are diagnosed in both men and women worldwide [6,7]. It is characterized by out-of-control growth in the colon or rectum or cecum. CRC is a genetic ailment in which the accumulated genetic changes in the colon confer a cell from adenomatous polyps with the malignant properties of out-of-control growth, the ability to invade neighboring tissues, resulting in metastasis [8]. Despite advances in prognosis and treatment, CRC remains a disorder with excessive morbidity and mortality [9]. The detection and treatment of early-degree CRC with the right agent serve a crucial function in decreasing the range of CRC [10]. Gefitinib (GFT) is a tyrosine kinase inhibitor drug (Figure 1a) used as a first-line treatment for patients with advanced or metastatic non-small cell lung, colon, and breast cancer [11,12]. GFT is a member of the biopharmaceutics classification system (BCS) class II drugs, which exhibit low solubility and high permeability after oral administration [13,14]. GFT shows poor solubility/absorption/bioavailability after oral administration with a dose of 250 mg once daily [15]. GFT is a dibasic compound with pKa values of 5.4 and 7.2, which shows a pH-dependent solubility in the gastrointestinal fluids (~60% of the drug is absorbed in the gastrointestinal tract (GIT) [16]. Several systems have been developed for enhancing the solubility of hydrophobic drugs, including medication derivatization, the use of a complexing agent, manipulation of the solid state, the use of a surface-active agent, enlarging the surface area of the drug exposed to dissolution, spray drying, and microencapsulation [14,17,18,19,20,21]. Recently, the use of lipid vesicular systems such as ethosomes, niosomes, liposomes, proliposomes, and cubosomes has been developed to enhance the solubility of poorly water-soluble drugs [22]; these systems not only enhance the solubility of poorly soluble drugs but also sustain the release rate, which enhances the bioavailability for many drugs [23,24,25,26].

Cubosomes are considered a potential carrier in vaccine and drug delivery. Cubosomes are biocompatible, bioadhesive nano-structured cubic systems obtained by colloid dispersion of a cubic liquid crystalline structure in water using a suitable surfactant [27]. They are composed of emulsification of biocompatible biodegradable lipids such as soy phosphatidylcholine, glyceryl oleate, and glyceryl monooleate (GMO) (Figure 1b) in water using stabilizers or surfactants, such as poloxamer 407 (P407) (Figure 1c) or Tween 80 [28].

Cubosomes offer many advantages over simple liposomes, such as their higher stability and mechanical rigidity. They also promote targeting and controlling drug release and entrapping a wide variety of drugs [29]. Due to their properties, they can be administered through different routes of administration, e.g., parenteral, oral, and percutaneous routes [30]. Cubosomes have attracted particular attention for improving the oral delivery of a variety of drugs, including high molecular weight and poorly soluble drugs [31]. Due to their bioadhesive features, the production of physiological surfactants in the GIT, and interactions with the intestinal cell membrane, cubosomes can aid in the absorption of drugs given orally [32]. Additionally, they can safeguard drugs from deterioration and have a great loading capacity [32]. Cubosomes are a useful option for improving the oral absorption of drugs that are not easily soluble [33]. By trapping the drug inside the mixed micelles that are created during cubosome digestion, cubosomes in the GIT maintain the drugs in the solubilized state. They thereby increase drug absorption, which increases oral bioavailability [34]. We used cubosomal nanoparticles (CNPs) as a drug delivery vehicle to assess the anti-cancer effects of gefitinib in this study. As far as we are aware, this is the first study to describe the creation of GFT-CNPs that may be taken orally and used to treat colon cancer. The synthesized GFT-CNP size, morphology, FTIR, DSC, and in vitro release were all measured. Additionally, the gene expression levels of TIMP-1 and MMP-7 by RT-PCR, estimation of colon biomarkers as serum cancer embryonic antigen (CEA) and carbohydrate antigen 19-9 (CA 19-9), and histopathological examination of colon tissues were estimated.

## 2. Materials and Methods

### 2.1. Materials

Beijing Mesochem Technology, Beijing, China, supplied the gefitinib (purity: 98.60%). GMO was obtained from Sigma Chemical Co., St. Louis, MO, USA. P407 was supplied by BASF Co., Ludwigshafen, Germany. Dimethylhydrazine (DMH) was obtained from Sigma-Aldrich (St. Louis, MO, USA). Other chemicals and reagents were of analytical quality and purchased from reliable sources.

### 2.2. Animals and Ethical Approval

Male Wistar rats that had reached sexual maturity were collected from the animal colony at the Faculty of Medicine, University of Mansoura, Egypt (150 g each). Every animal had unrestricted access to a standard grain feed and running water in a sterile environment (22 ± 2 °C temperature, 50 ± 5% humidity, 12 h light/12 h dark cycle). Care and exploitation of the animals conformed to the National Institutes of Health Guide for the Care and Use of Laboratory Animals and was sanctioned by the Animal Ethics Committee of the Faculty of Pharmacy at Al-Azhar University in Assuit, Egypt (ZA-AS/PH/26/C/2022).

### 2.3. Methods

#### 2.3.1. Preparation of Gefitinib Cubosomal Nanoparticles (GFT-CNPs)

GFT-CNPs were prepared by the emulsification of monoglyceride by the surfactant system in water [35]. The dispersed phase (in 3 ratios) composed of GMO and P407 was melted in a thermostatically controlled water bath at 70 °C until a homogeneous mixture was obtained. Then, GFT was uniformly mixed with the molten mixture. The GFT/GMO/P407 molten dispersion was introduced to H_2_O dropwise at the same temperature, with continuous stirring at 1500 rpm by a wisestir magnetic stirrer (Ms-300HS, misung scientific Co., Korea). The obtained coarse dispersion was stored for 24 h at room temperature till equilibrium, then subjected to ultrasonication utilizing an ultrasonic processor, GE130, probe CV18 [36]. This dispersion was then homogenized for 2 h at 10,000 rpm using a Heidolph silent crusher^®^ homogenizer, Palo Alto, München, Germany.

#### 2.3.2. Experimental Design

A full factorial (3^2^) experimental design was constructed using Stratigraphic Plus^®^ 18 software, Stat point Tech., Inc. Warrenton, VA, USA [37,38,39,40]. Three levels of each independent variable were used for the optimization of the prepared GFT-CNP formulations [41]. The selected levels of GMO/P407 concentration in total dispersion (X_1_) were 2.5 (−1), 5 (0), and 7.5 (+1)% (*w*/*w*), whereas the selected values for P407 concentration in GMO/P407 mixture (X_2_) were 5 (−1), 10 (0), and 15 (+1) % (*w*/*w*) (Table 1). Entrapment efficient percentage (EE%) (Y_1_), particle size (Y_2_), and cumulative percent released at 12 h (Y_3_) were selected as dependent (response) parameters.

#### 2.3.3. Characterization of Prepared GFT-CNPs

##### Estimation of Entrapment Efficiency Percentage (EE%)

Before the estimation of the EE%, the drug aggregates were separated by simple filtration [42]. GFT-CNPs were separated from un-entrapped GFT by centrifugation of a known volume of cubosomal dispersion (3 mL) for 1 h at 15,000 rpm using a cooling centrifuge at 4 °C. After separation of the GFT-CNPs from the supernatant, they were washed two times with sufficient saline solution (0.9% *w*/*v*) until the absence of un-entrapped GFT from the voids separating GFT-CNPs and centrifugated again for 30 min. The entrapped GFT was estimated after the destruction of the vesicles by sonication with methanol for 10 min using the sonication digital sonifier (Branson, Danbury, CT, USA) [43]. The concentration of the entrapped GFT was measured spectrophotometrically utilizing Shimadzu, model UV-1601 PC, Kyoto, Japan at an absorption maximum (λ_max_) of 249.5 nm. All procedures were performed in triplicate, and the mean value was considered ± SD. The EE% was determined according to Equation (1).
EE% = [(Ct – Cf)/Ct] × 100(1)
where Ct is the total amount of GFT added, and Cf is the amount of free GFT

##### Morphology of the Prepared GFT-CNPs

GFT-CNPs GC8 and GC9 (plain and medicated) were examined by transmission electron microscopy TEM, (JEM-100 CX, Osaka, Japan) at 80 kV. First, the sample was diluted with distilled water and sonicated for 5 min. Then, drops from the diluted dispersion were spread over a copper-coated grid, followed by staining with a phosphotungstic acid solution, and were dried by air [44,45].

##### GFT-CNP Size and Polydispersity Index (PDI)

Malvern Zetasizer 300 HAS (Malvern Instrument, Malvern, UK) was used to determine GFT-CNP size and PDI. Distilled water was used to dilute GFT-CNPs to supply an appropriate scattering intensity. Samples were placed in square glass cuvettes and inspected at 25 ± 0.5 °C three times, and the average was listed ± SD. PDI indicates the homogeneity of the particle size [5].

##### Zeta Potential Determination

Photon correlation spectroscopy (Zetasizer Nano ZS, ZEN 3600; Malvern Instruments, Malvern, UK) was utilized to calculate the Zeta potential of GFT-CNPs. The investigated samples were suitably diluted with distilled water and placed in clear disposable zeta cells. Six measurements were taken, and the average was listed ± SD [46].

##### Differential Scanning Calorimetry (DSC)

DSC (Shimadzu DSC-50) equipped with an intercooler 1 p was used to detect the physical state of GFT and the possible interaction between GFT and other excipients in the formulation. GMO, P407, GFT, and optimized GFT-CNP (GC9) samples (5 mg) were placed in an aluminum pan and scanned at a rate of 10 °C/min over a temperature range of 15–400 °C under a purge of nitrogen. An identical empty pan was used as a reference. Thermograms were analyzed using a Shimadzu TA-60 software system (Lbsolutions TA workstation) [47]. The various obtained thermograms were evaluated for peak shift as well as appearance and disappearance of new peaks.

##### Fourier Transform Infrared (FTIR) Spectroscopy

To detect any possible interaction between the drug and other formulation ingredients, an FTIR spectroscopy investigation was performed. GMO, P407, GFT, and optimized GFT-CNPs (GC9) were investigated by Fourier Transform (IR-476-Shimadzu Kyoto, Japan) employing the KBr disc technique between 4000 and 400 cm^−1^. The samples were compressed at a pressure of 6 tons/cm^2^ using a Shimadzu SSP-CoA IR compression machine following trituration and thorough mixing with KBr. The spectra were acquired at 2 cm^−1^ after the plates were charged in the instrument’s light path [48].

#### 2.3.4. In Vitro GFT-CNPs Release and Kinetic Studies

A horizontal shaking water bath (Gesellschatt laboratories, Berlin, Germany) was used for the determination of the in vitro release of GFT from the formulated cubosomes. GFT-CNP dispersion equivalent to 5 mg GFT was introduced into a dialysis bag (semipermeable cellophane membrane having 12,000 molecular weight cut-off range and tied from both ends with cotton threads) and immersed in a beaker containing 100 mL phosphate buffer, pH 7.4. The system was maintained at 37 ± 0.5 °C. At a specified time, at an interval of 12 h, 1 mL of the dialysate was withdrawn and replaced with the same volume of fresh phosphate buffer, pH7.4, to maintain the sink condition. The amount of GFT released was measured spectrophotometrically at 249.5 nm using phosphate buffer, pH7.4, as blank. The listed results are the mean values of the release experiments (*n* = 3) [49]. GFT-CNP release profiles were drawn by plotting the cumulative percentage of drugs released at each time interval against time. Pure GFT suspension (as a control) release pattern was compared to the release pattern of different GFT-CNPs formulations.

The amount of GFT released at each time interval was calculated according to Equation (2).
(2)$$\mathrm{GFT}\;\mathrm{amount}\;\mathrm{released}\;(\mathrm{mg})=\frac{\mathrm{Abs}.\times P.C.\times D.F.\times \mathrm{Vr}}{1000}$$
where Abs.: UV absorbance at every time point of sampling; P.C.: procedural constant obtained from GFT standard curve; D.F.: dilution factor; and Vr: receptor medium volume.

To determine which kinetics model (zero-order, first-order, or Higuchi-diffusion) best fit the observed in vitro release data, the following procedures were carried out:

Zero order rate equation [50]:Q_t_ = Q_0_ + K_0_t(3)
where Q_t_ is the amount of drug released at time t, Q_0_ is an initial amount of drug, and K_0_ is the zero order release rate constant.

The zero order release rate constant, K_0_, is related to the starting drug quantity, Q_0_, and the amount of drug release at time t, Q_t_.

First order rate equation [51]:logQ = logQ_0_ − K_1_t/2.303(4)
where Q is the remaining amount of drug at time t, Q_0_ is the initial amount of drug, K_1_ is the first order release rate constant, and t is time.

Higuchi’s model [52]:Q = K_H_t^1/2^(5)
where Q is the amount of drug released in time t per unit area at time t, and K_H_ is the Higuchi diffusion rate constant.

The best-fitted model was selected based on the regression coefficient (R^2^) value.

Korsmeyer-Peppas model [53]:

To determine the mechanism of drug release, the in vitro release data of all cubosomal dispersions were fitted to the Korsmeyer-Peppas equation.
M_t_/M_∞_ = Kt^n^(6)
where M_t_/M_∞_ is a fraction of drug release at time t, K is the release rate constant, and n is the release exponent. The value of exponent (n) indicates the mechanism of drug release.

#### 2.3.5. Selection of the Optimum Formula

The best GFT-CNPs formulations were chosen according to the highest EE%, the lowest vesicle size, and cumulative % released after 12 h. Depending on the obtained data, F9 (composed of 7.5% *w*/*w* GMO/P407 and 15% P407) was selected as the best one and used for further investigation.

#### 2.3.6. GFT-CNP Stability Study

The optimized GFT-CNP formula (GC9) was placed in vials (borosilicate, clean, dry, airproof, and dark-colored) and stored at 4 ± 0.5 °C and 25 ± 2 °C for 6 months [54]. The size of the vesicle and EE% were investigated and compared to the same properties of the stored formulation at time zero. Each test was conducted in a triplicate manner, and the average was listed ± SD [55].

#### 2.3.7. In Vivo Study

##### Animal Experiments

After a two-week adaptation period, 90 healthy male rats were randomly categorized into six groups, with 15 animals in each group. Animals in groups 1 and 2 were not exposed to any carcinogenic material (DMH) and were provided with sodium carboxymethyl cellulose (CMC-Na) in water (0.5% *w*/*v*) and blank cubosomes orally every day during the test period. Animals in groups 3, 4, 5, and 6 received a single subcutaneous injection of DMH (20 mg/kg) twice weekly for four consecutive weeks to promote the carcinogenic effect of DMH [56]. Two weeks after the last DMH dose, the positive control groups (3 and 4) received a solution of CMC-Na in water (0.5% *w*/*v*) and blank cubosomes, individually every day by oral gavage during the test. Animals in group 5 received a dose of (10 mg/kg body weight per day) for six weeks from GFT suspended in a solution of CMC-Na in water (0.5% *w*/*v*) by oral gavage [57]. Animals in group 6 received the same dose amount (10 mg/kg body weight per day) for six weeks from GFT-CNPs, as shown in group 5.

##### Serum Specimen Collection

After a 12 h fasting interval following the last dosage, blood samples were drawn through heart puncture while the animals were under ketamine anesthesia (100 mg/kg/intraperitoneally). After letting the blood samples clot for 20 min, they were centrifuged at 4000 rpm for 15 min at 4 °C in a chilled centrifuge (Beckman model L3-50, USA), and the serum was collected and frozen at −80 °C.

##### Tissue Specimen

Cervical dislocation was used to put the animals to sleep. Then, we separated the colon, washed it using a phosphate-buffered saline solution kept at a cool temperature (PBS), patted it dry using filter paper, and cut it in half. The initial sample was frozen at −80 °C for later real-time polymerase chain reaction analysis (PCR). Histopathological analysis was performed on the second segment after it was preserved in 10% formalin.

##### Detection of TIMP-1 and MMP-7 by Real-Time PCR

The Kit for the Isolation of Total RNA from GF-1 (Vivantis Technologies, China) was used to extract RNA from certain subsections of the colon, and the lysis solution included in the kit was used to provide the maximum recovery of functional RNA. The entire process, including the isolation of total RNA, was carried out with the aid of reagents kept at extremely low temperatures by being placed on ice. A SPECTROstar Nano spectrometer was used to quantify RNA concentrations (BMG Labtech, Germany). The 260/280 ratio was used as an indicator of RNA purity. As instructed by the manufacturer, 2 g of total RNA was converted into single-stranded cDNA using the cDNA synthesis kit included in a 2-step RT-PCR kit (Vivantis Technologies). Using a total reaction volume of 20 L per well on an RT-PCR plate, we measured target mRNA expression using real-time (RT)-PCR amplification (Applied Biosystem step one, France). Two microliters of cDNA, 33 microliters of each 10 mM primer, 10 microliters of SYBR Green universal master mix (Thermo Fisher Scientific, USA), and 7.5 microliters of DNase-free water were used to initiate the amplification reaction. The following describes the RT-PCR reaction conditions: there were 35 iterations of denaturation (95 °C for 3 min), annealing (at a temperature determined by a particular gene for 30 s), and extension (72 °C for 30 s) before the final extension (72 °C for 5 min). The primers were created with the help of PubMed and tested at various annealing temperatures before being purchased from Vivantis Technologies (Malaysia).

TIMP-1 forward Tm = 59.10 °C: 5′-CAGCAGTGGGTGGATGAGTA-3′

TIMP-1 reverse Tm = 59.89 °C: 5′-AGCAGGGCTCAGATTATGCC-3′

MMP-7 forward Tm = 60.61 °C: 5′-CTCACCCTGTTCCGCATTGT-3′

MMP-7 reverse Tm = 59.67 °C: 5′-TCCCCTTGCGAAGCCAATTA-3′

β-actin forward Tm = 61.01 °C: 5′-CACCCGCGAGTACAACCTTC-3′

β-actin reverse Tm = 59.96 °C: 5′-GTACATGGCTGGGGTGTTGA-3′

The RT-PCR data were evaluated to calculate fold changes and relative expression using the 2^−△△Ct^ method by Livak [58]. β-actin was used as the endogenous reference gene.

##### Estimation of Colon Biomarkers

Serum cancer embryonic antigen (CEA) and carbohydrate antigen 19-9 (CA 19-9) levels were determined using an enzyme-linked immunosorbent assay kit from Glory Science (Hangzhou, China).

##### Histopathological Assessment

Sections of the specimens were cut at a thickness of 4–5 µm and examined for tissue alterations in the colon after they were fixed in 10% formalin and dried in a series of ethanol washes. A histopathologist who was blinded to the treatment details evaluated the hematoxylin and eosin (H&E)-stained sections using a light microscope (Olympus, Center Valley, PA, USA) [59].

##### Statistical Analysis

GraphPad Prism, version 5.0, was used for statistical analysis (GraphPad, San Diego, CA, USA). Statistical analysis of variance followed by Tukey’s t-test was used to evaluate the differences between groups. All findings were plotted as means plus standard error of the mean, and a *p* value of 0.05 met the criteria for statistical significance.

## 3. Results and Discussion

### 3.1. Fabrication of GFT-CNPs

The GFT-loaded cubosomal nanoparticles (GFT-CNPs) were effectively fabricated by disrupting a cubic gel phase of GMO and distilled water in the presence of a stabilizer (P407) with the aid of mechanical stirring ad ultrasonication. Nine formulations were used, and the composition of the investigated cubosomal formulations is listed in Table 2. The fabricated formulations were evaluated visually as a homogenous gel (opalescent dispersion of the cubic nanoparticles). Since GMO has double bonds (unsaturation), this may help its interaction with the drug, preventing phase separation. Entrapment efficiency percentage (EE%) (Y_1_), GFT-CNPs size (Y_2_), and the cumulative amount released at 12 h (Y_3_) were selected as the response variables.

### 3.2. DSC Studies

DSC was carried out to determine the crystalline properties of the loaded GFT in optimized cubosomal nanoparticles compared with pure GFT, P407, and GMO. Figure 2 shows the DSC thermograms of pure GFT, GMO, P407, and optimized GFT-CNP formulation (GC9). Pure GFT showed a sharp endothermic melting peak at 198.54 °C, which corresponds to its melting point (Figure 2a). Following the literature, the DSC thermogram obtained with GMO demonstrated a broad peak at 39.97 °C due to its melting points (Figure 2b). The DSC thermogram P407 revealed two endothermic transitions around 52.3 °C, corresponding to the melting temperature of P407 (Figure 2c). The optimized GFT-CNPs revealed an endothermic peak at 37.12 °C, showing slight shifts of the GMO endothermic peak (Figure 2d). These shifts could be attributed to the development of the bicontinuous structure between water and GMOs [32]. The thermograms of the GFT-CNPs also revealed the disappearance of the sharp peak of GFT, revealing its alteration to the amorphous state and signifying that GFT was incorporated in the bicontinuous cubosomal structure [60].

### 3.3. FTIR Spectroscopy

The chemical compatibility was ensured by carrying out FTIR spectral analysis of the drug with excipients and by comparing the peaks (Figure 3). The FTIR analysis was performed to support the results obtained from the DSC thermal analysis. The FTIR spectra for pure GFT exhibited various spectral peaks for the existing functional groups. The functional group NH showed a sharp peak at 3400.8 cm^−^^1^. The other peaks were observed at 2956.79 cm^−^^1^ (CH2), 1625.08 cm^−^^1^ (C=N), and 1578.39 cm^−^^1^ (C=C of the aryl group). Pure GFT also exhibited peaks for the C-O group at 1110.18 cm^−^^1^. The presence of a peak at 930.31 cm^−^^1^ further revealed the presence of a C-F group, which is in alignment with the chemical structure of pure GFT. The FTIR spectra of GMO demonstrated an OH stretching peak at 3442.35 cm^−1^. The CH_2_ stretching and ester bond were definite by peaks at 2924.33 cm^−1^ and 1735.96 cm^−1^, respectively. The FTIR spectrum of P407 showed a characteristic broadband at 3450.76 cm^−1^ for the O–H group. Peaks at 2970 and 2988.53 cm^−1^ represent the C–H group. In addition, a peak appeared at 1111.91 cm^−1^, representing stretching C–O. The FTIR spectra of the optimized GFT-CNP formula (GC9) showed a slight shift, decreased peak intensity, and disappearance of some GFT characteristic peaks. This result is in agreement with the DSC thermograms and is attributed to the possible interaction between GMO and GFT [61]. Based on the results of DSC thermograms and FTIR spectra, it is concluded that GMO showed a remarkable affinity to interact with GFT.

### 3.4. GFT-CNPs Morphology

The TEM microphotographs of the optimized GFT-CNPs (GC9), plain and medicated, were compared to the GFT-CNP formula, GC8 (Figure 4). Figure 4 reveals that the investigated formulations were cubic. It has been previously reported that utilizing P407 in nanosized colloidal vesicles can alter their spherical shape into a cubic shape [62]. Moreover, the investigated cubic particle diameters determined by TEM were noticeably smaller than those determined by the Zeta sizer during particle size determination. The size reduction could be due to the dehydration of the vesicular system due to drying during the TEM investigations.

### 3.5. Response Surface Methodology and Optimization of Formulation Factors

#### 3.5.1. Entrapment Efficiency Percentage (EE%)

The EE% of all GFT-CNPs formulations are represented in Figure 5. The EE% was in the range between 45.24 ± 3.87 and 83.11 ± 4.39. The main effects plot and response surface figure (Figure 6) showed that the GFT-CNPs containing a higher concentration of P407 have higher EE% than those containing lower concentrations. The EE% was found to be affected by the concentration of P407 in addition to the lipophilicity of the GFT. As the concentration of P407 increased, we observed an increase in EE%. When the content of poloxamer 407 increased, the ability of the formulated CNPs to hold GFT increased, possibly due to the increased hydrophilicity of the GFT-CNPs and entrapment of GFT into the aqueous core together with P407. The obtained results were in agreement with many studies showing that P407 (a surfactant with low hydrophilic-lipophilic balance) is the stabilizer of choice in cubosome preparation [32].

#### 3.5.2. Particle Size, PDI, and Zeta Potential

The obtained results of the fabricated GFT-CNP formulations’ mean particle size and PDI are presented in Table 2. The mean particle size of the GFT-CNP dispersions was in the range between 98 ± 20 (GC3) and 550 ± 11 nm (GC7). The obtained results revealed that there was a strong relationship between both the GMO/P407 ratio and P407 concentration and particle size of the investigated GFT-CNPs (Figure 7). At a low GMO/P407 ratio (2.5%) concerning the final dispersion volume, a large particle size was obtained [63]. As the GMO/P407 ratio was increased (5% in GC2, 5, and 8), the particle size was decreased and further decreased as the GMO/P407 ratio was increased to 7.5% [60]. Moreover, the concentration of the stabilizer (P407) significantly affected the particle size. The obtained results showed that as the P407 concentration increased, the cubosomal particle size increased [64]. The PDI of the dispersions was in the range between 0.257 ± 0.04 and 0.493 ± 0.05. The obtained PDI values (from 0.132 ± 0.052 to 0.352 ± 0.029) are considered acceptable for lipid-based carriers’ drug delivery systems and indicate the homogeneity of the particle size in the formulated GFT-CNP dispersions, with low affinity to aggregation [63]. The particle size distribution curve of GC9 is illustrated in Figure 8. The results of the zeta potential for the nine GFT-CPN dispersions were in the range between −25 ± 1.3 and −36 ± 0.68 mV (Table 2 and Figure 9). For this reason, it was proposed that the aforementioned nine formulations of GFT-CNPs have very high stability. It was previously reported that nanoparticles with zeta potential values higher than −25 mV possess a higher stability degree than nanoparticle dispersions with a low zeta potential value due to Van Der Waals inter-particle attractions (have a higher tendency to aggregate) [65].

#### 3.5.3. In Vitro GFT-CNP Release and Kinetic Studies

The in vitro release data indicate how the drug will behave in vivo. Using a control GFT-suspension formulation and these few alternative GFT-CNP formulations, in vitro release tests looked at the total amount of GFT released over time (Figure 10). The data indicated that the GFT-suspension formulation had a typical half-life of 2 h before detectable drug concentrations arose (control). In addition, just 34.51 ± 4.65 percent of the total GFT was released after 12 h. Since GFT is only slightly soluble in water, the medication had to be dissolved from progressively bigger particle sizes before it could diffuse into the donor compartment over the cellophane membrane. Cubosomes have a bi-phasic release pattern, where initial burst release occurs within the first 2 h, followed by a sustained release pattern over 12 h. The initial release may be due to the diffusion of the un-entrapped GFT, which could be easily diffused to the matrix, while the sustained release pattern may be attributed to the diffusion of GFT through the water channels within the cubic nanoparticles. For GFT-CNP formulations, the cumulative percentages of GFT released from the investigated cubosomes ranged from 45.37 ± 4.21% (GC7) to 88.06 ± 3.78% (GC3) after 12 h (Figure 10). These results indicated a significant improvement (*p* < 0.005) in the cumulative percentages of GFT release, compared to the control GFT-suspension formulation. This significant enhancement in % GFT release could be ascribed to the nanonization of cubosomes and the availability of GFT molecules in the soluble form within the lipid bilayer domains of the cubosomes. Drug molecules required only partition throughout the lipid bilayer to the bulk of the phosphate buffer, pH 7.4 release medium [66]. Generally, increasing stabilizer (P407) ratios in cubosomes resulted in a significant decrease (*p* < 0.0001) in the cumulative percentages of GFT released (controlled release) (Figure 11). This could be because increasing the concentrations of stabilizers led to increasing the affinity of GFT to the cubosomes and increasing its solubilization in their tortuous aqueous domains [66]. Additionally, increasing the stabilizers’ concentrations is likely to allow more stabilizer molecules to be accommodated within the formed cubosomes and consequently could delay the release of GFT from the fabricated cubosomal formulations [67].

The obtained results showed that the larger the particle size of cubosomes, the smaller the cubosomal surface area exposed to the release media, which results in a sustained drug release [68]. In particular, GC3, prepared with lower P407 content, showed a prompt drug release and significantly (*p* < 0.05) higher value of cumulative percent released after 12 h as compared to pure GFT suspension and the other cubosomal formulations. This may be attributed to the extremely small particle size of GC3 (98 ± 20 nm), which provides a larger effective surface area for drug release and a short diffusion distance for the rapid release of the entrapped drug from cubosomes. A rapid release of lipophilic drugs from cubosomes has been previously reported [69]. The best-fit model for GFT release from the fabricated nanosized cubosomes was Korsemeyer-Peppas, with the highest R^2^ (Table 3). The diffusion exponent was between 0.5 and 1, and the GFT release from GFT-CNP dispersions underwent a non-fickian release mechanism, revealing that the release was controlled by the GFT diffusion rate and the relaxation rate of the polymer (P407).

### 3.6. GFT-CNP Stability Study

The obtained results of the particle size and EE% of GC9 over 6 months are reported in Figure 12. When the GFT-CNP GC9 formula was stored at room temperature (25 ± 2 °C) over 6 months, a significant (*p* < 0.05) increase in particle size from 365 ± 20 nm to 504 ± 18 nm was noticed. However, upon storage at refrigerated conditions (4 ± 0.5 °C), GC9 remained stable without any significant change in the particle size after 6 months (370 ± 15 nm). The presence of P407 present in the GC9 cubosomal formulation could have stabilized the lipid structures by preventing its aggregation during the storage period [70]. At room temperature, the EE% of GC9 was also reduced from 83.11 ± 4.39% to 46.11 ± 3.3%. During storage of GC9 at refrigerated conditions, an insignificant reduction in the EE% of the cubosomes was observed (Figure 12). At 25 ± 2 °C, GC9 cubosomal formulation could have acquired energy from the surrounding heat and light, which enhance the Brownian motion of cubosomal particles. This behavior promotes the collision of cubosomal particles with each other, resulting in P407 coating removal. The depletion of the P407 coating could lead to the adhesion of cubosomal particles to one another to generate deformed cubosomal particles with bigger sizes and reduced EE% [70]. Thus, storage of GC9 cubosomal formulation at refrigerated conditions is a suggested option to avoid aggregation and drug leakage during its storage.

### 3.7. In Vivo Study

#### 3.7.1. Gene Expression Levels of TIMP-1 and MMP-7 by RT-PCR

Colon TIMP-1 and MMP-7 gene expression were significantly increased in the carcinogenic group compared with the control group (*p* < 0.001). Whereas, there was a significant decrease in these genes in the group treated with GFT-CNPs compared with the carcinogenic group (*p* < 0.001) (Figure 13a,b). This data agreed with Meng et al. that revealed that TIMP-1 has potential diagnostic value with upper moderate sensitivity and specificity, and TIMP-1 measurement might be useful as a noninvasive screening tool for the clinical practice of colon cancer [71]. The results also agreed with the study that reported that MMP-7-expression is a prognostic marker for poor 5-year outcomes in colorectal cancer and is a potential target for tumor therapy [72]. In the same context, tissue inhibitors of matrix metalloproteinases (TIMPs), naturally occurring tissue inhibitors of matrix metalloproteinases (MMPs), partly regulate the proteolytic activity of MMPs, stimulating tumor growth and inhibiting tumor cell apoptosis, and also act as a functional regulator of malignant transformation [71].

#### 3.7.2. Serum Levels of CEA and CA19-9

Our assessment demonstrated that the serum levels of CEA and CA19-9 were significantly increased in the carcinogenic group compared with the control group (*p* < 0.001). Furthermore, the treatment with GFT-CNPs led to a significant decrease in the serum level of CEA and CA19-9 compared with the carcinogenic group (*p* < 0.001) (Figure 13c,d). These results are in agreement with a previous study that reported that CEA and CA 19-9 are the most common tumor-associated antigens used in the staging of patients with rectal cancer and in other parts of the colon [73,74].

#### 3.7.3. Histopathological Examination

The examined colon tissues’ histopathological characteristics showed that control groups had colon tissue with no histopathological alteration, while the diseased groups had focal ulceration in the lining mucosal epithelium associated with inflammatory cell infiltration in the underlying lamina propria. The focal ulceration and inflammatory cell infiltration were still observed after treatment with GFT suspension. However, after treatment with GFT-CNPs, there was an enhancement and a decrease in the focal inflammatory cell infiltration in the lamina propria, as shown in Figure 14.

## 4. Conclusions

GFT-CNPs were successfully prepared by the emulsification method. The GMO/P407 ratio and P407 concentration strongly affect EE%, vesicle size, and in vitro GFT release. GFT-CNPs provide a sustained GFT release rate. GFT-CNPs had a semi-cubical shape in the nano-size range and had higher EE%. The zeta-potential values of the formulated GFT-CNPs indicated that the system is stable. GFT-CNP (GC9) was chosen as the optimum formula and exposed to several examinations, such as FTIR and DSC, which confirmed the incorporation of GFT in the bicontinuous cubosomal structure. Stability studies showed that when GFT-CNPs were stored at room temperature, there was a significant increase in the vesicle size and decrease in EE%, but when stored under refrigerated conditions, remained stable, without any significant change in vesicle size and an insignificant reduction in EE%. The in vivo data showed a significant decrease in the serum level of embryonic antigen (CEA), carbohydrate antigen 19-9 (CA 19-9), and gene expression level of TIMP-1 and MMP-7. Histopathological examination showed enhancement in cancer tissue and highly decreased focal infiltration in the lamina propria after treatment with GFT-CNPs. This study firmly suggests the possible usage of GFT-CNPs as an oral vesicular system for the treatment of colon cancer.

## Figures and Tables

**Figure 1 pharmaceutics-15-00680-f001:**
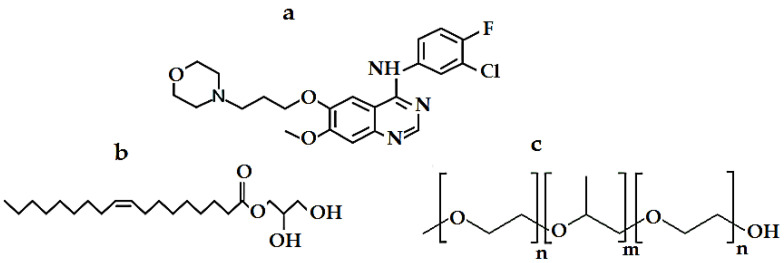
Chemical structure of (**a**) Gefitinib, (**b**) Glyceryl monooleate, and (**c**) Poloxamer 407.

**Figure 2 pharmaceutics-15-00680-f002:**
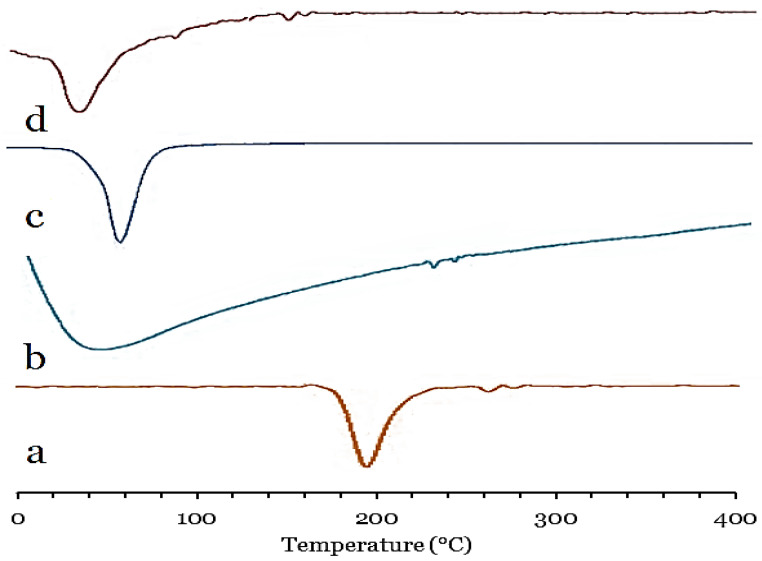
DSC thermograms of (**a**) pure GFT, (**b**) GMO, (**c**) P407, and (**d**) optimized GFT-CNP formulation (GC9).

**Figure 3 pharmaceutics-15-00680-f003:**
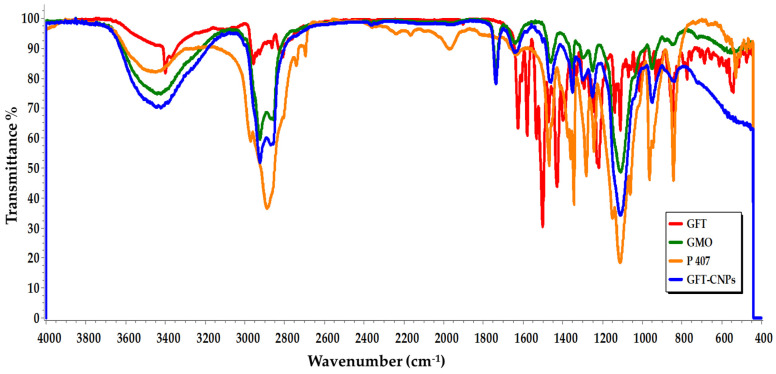
FTIR spectra of pure GFT, GMO, P407, and optimized GFT-CNP formula (GC9).

**Figure 4 pharmaceutics-15-00680-f004:**
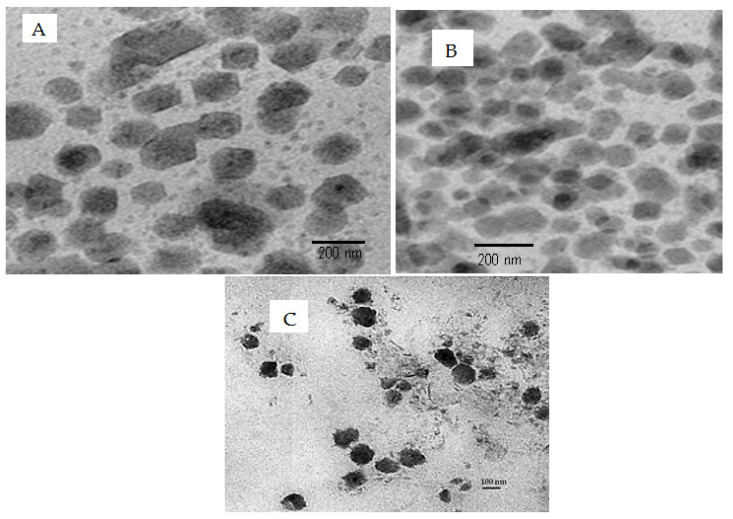
TEM micrograph of (**A**) GFT-CNP plain optimized formulation (GC9), (**B**) GFT-CNP, GC8, and (**C**) GFT-CNP optimized formulation (GC9).

**Figure 5 pharmaceutics-15-00680-f005:**
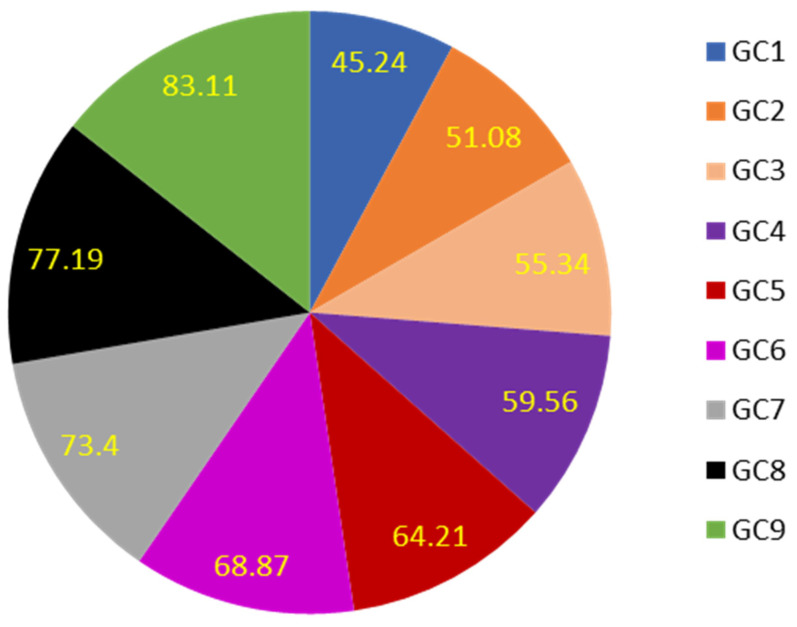
EE% of GFT-CNP formulae (GC1–GC9).

**Figure 6 pharmaceutics-15-00680-f006:**
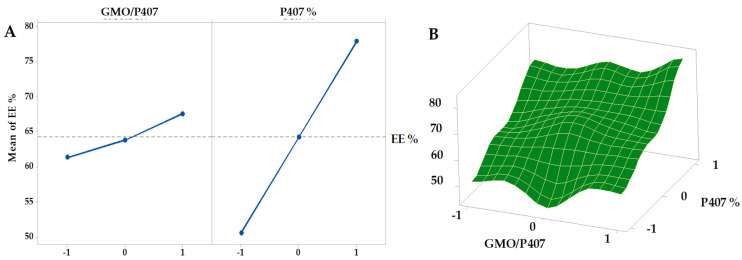
Main effects (**A**) and response surface plots (**B**) of GMO/P407 and P407% on EE%.

**Figure 7 pharmaceutics-15-00680-f007:**
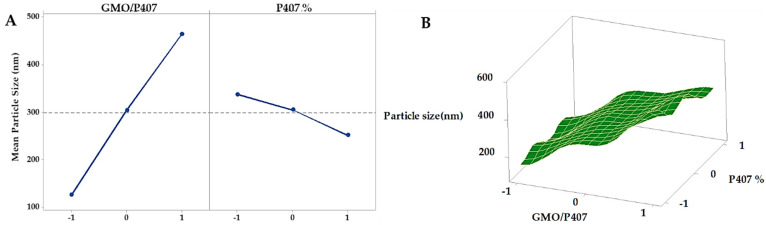
Main effects (**A**) and response surface plots (**B**) of GMO/P407 and P407% on particle size.

**Figure 8 pharmaceutics-15-00680-f008:**
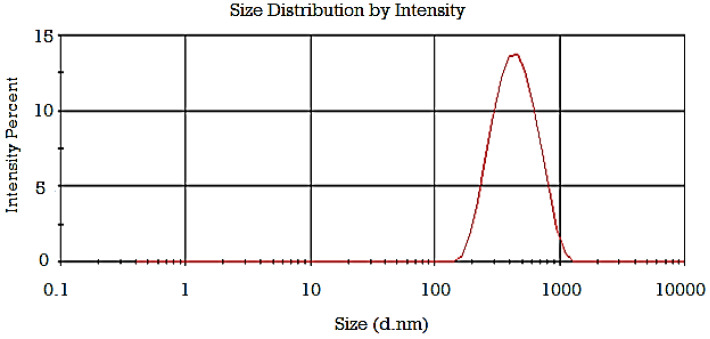
Particle size distribution of GFT-CNP formulation (GC9).

**Figure 9 pharmaceutics-15-00680-f009:**
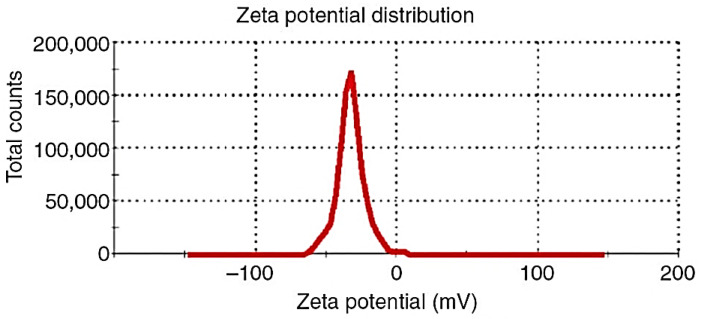
Zeta potential of GFT-CNP formulation (GC9).

**Figure 10 pharmaceutics-15-00680-f010:**
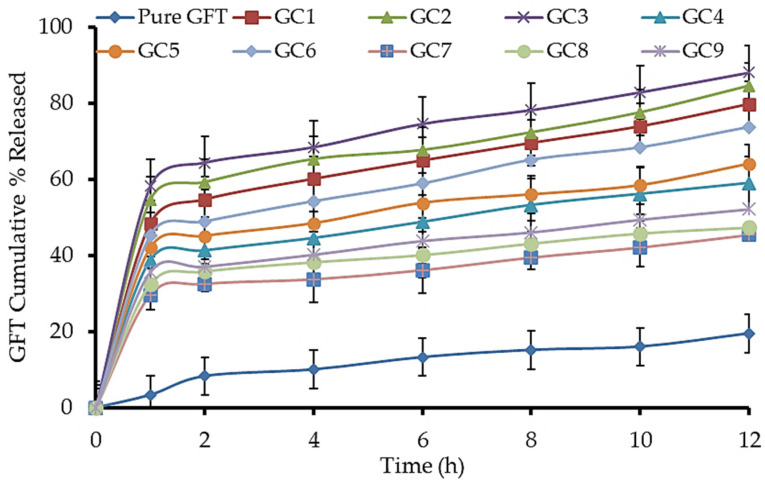
In vitro release profiles of pure GFT-suspension and GFT-CNP formulation (GC1–GC9) in phosphate buffer, pH7.4, at 37 ± 0.5 °C (*n* = 3; the data are expressed as mean ±SD).

**Figure 11 pharmaceutics-15-00680-f011:**
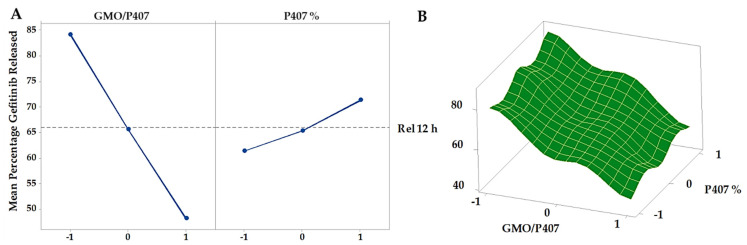
Main effects (**A**) and response surface plots (**B**) of GMO/P407 and P407% on cumulative percentage GFT released.

**Figure 12 pharmaceutics-15-00680-f012:**
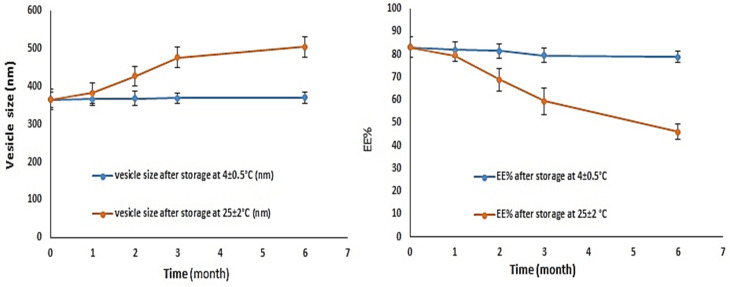
Stability profiles of optimum GFT-CNPs (GC9) under storage at 4 °C and 25 °C for 6 months (mean ± SD, *n* = 3).

**Figure 13 pharmaceutics-15-00680-f013:**
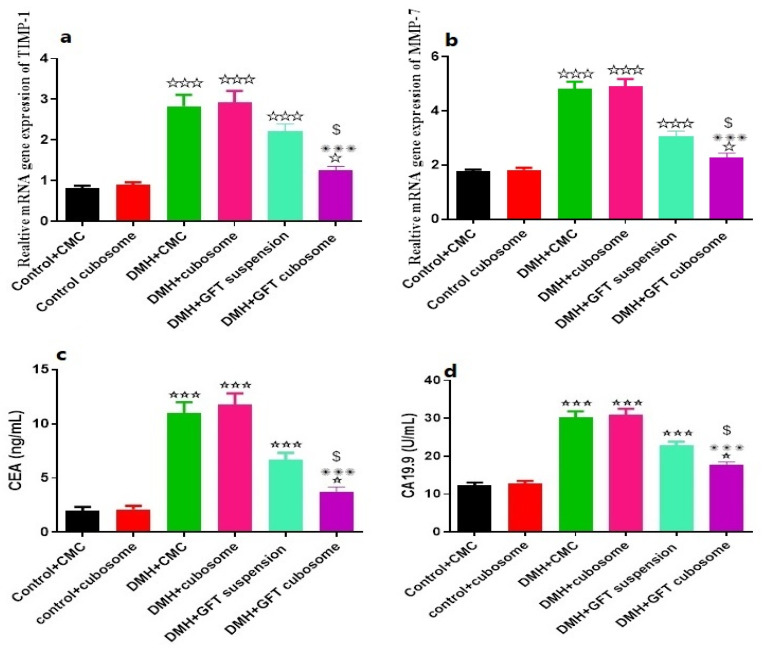
Effects of GFT and/or GFT-CNPs on gene expression levels of (**a**) TIMP-1 and (**b**) MMP-7, and serum levels of (**c**) CEA and (**d**) CA19-9. Data are presented as mean ± SEM (*n* = 12). ☆, ✺ and $ indicate significant change from Control group, DMH group and DMH + GFT suspension respectively. ☆ and $, indicate significant change at *p* < 0.05; ☆☆☆ and ✺✺✺ indicate significant change at *p* < 0.001.

**Figure 14 pharmaceutics-15-00680-f014:**
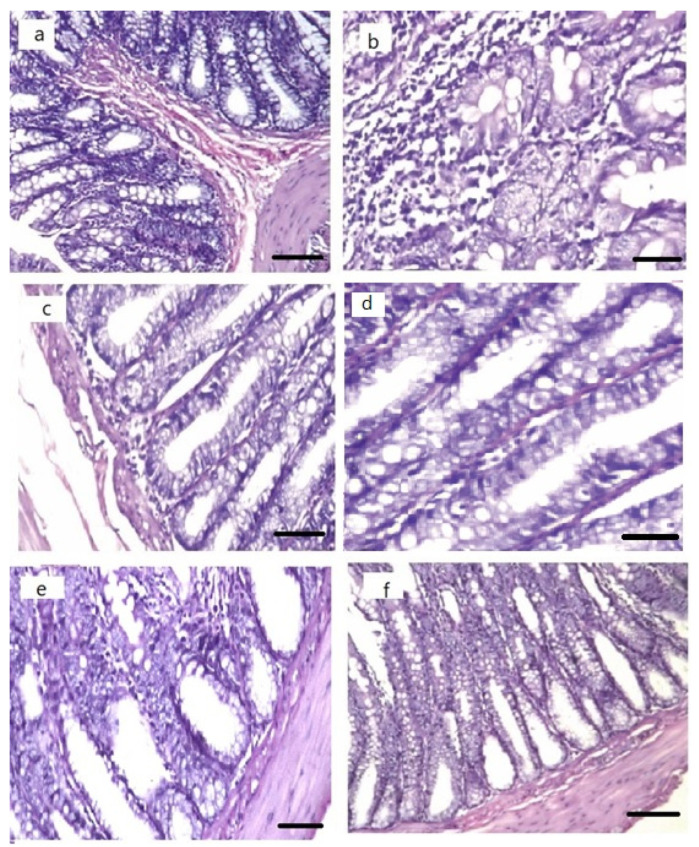
Histopathological changes of colon tissues: (**a**,**b**) colon tissues of control animals showing normal colon tissue with no histopathological alteration (H&E staining; scale bar, 200 µm), (**c**,**d**) colon tissues of diseased animals showing focal ulceration in the lining mucosal epithelium associated with inflammatory cell infiltration in the underlying lamina propria (H&E staining; scale bar, 200 µm), (**e**) colon tissues of diseased animals treated with GFT suspension showing focal inflammatory cell infiltration in the lamina propria (H&E staining; scale bar, 200 µm), (**f**) colon tissues of infected animals treated with GFT-CNPs showing enhancement and decrease in the focal inflammatory cell infiltration in the lamina propria (H&E staining; scale bar, 200 µm).

**Table 1 pharmaceutics-15-00680-t001:** Composition of different GFT-CNPs.

Composition	GC1	GC2	GC3	GC4	GC5	GC6	GC7	GC8	GC9
GFT (mg)	10	10	10	10	10	10	10	10	10
GMO/P407 (% *w*/*w*)	2.5	5	7.5	2.5	5	7.5	2.5	5	7.5
P407 (% *w*/*w*)	5	5	5	10	10	10	15	15	15
Water to (mL)	20	20	20	20	20	20	20	20	20

GMO/P407% level code: 2.5% (−1), 5% (0), and 7.5% (+1); P407% level code: 5% (−1), 10% (0), and 15% (+1).

**Table 2 pharmaceutics-15-00680-t002:** Particle size, PDI, and zeta potential and cumulative % released at 12 h of GFT-CNP formulations (values are represented as the mean ± SD).

F. Code	Particle Size (nm)	PDI	Zeta Potential (mV)	Cumulative % Released at 12 h
GC1	149 ± 24	0.157 ± 0.022	−29 ± 0.925	97.74 ± 1.43
GC2	132 ± 19	0.251 ± 0.052	−26 ± 0.790	84.56 ± 1.54
GC3	98± 20	0.302 ± 0.041	−27 ± 0.884	88.06 ± 2.01
GC4	312 ± 22	0.247 ± 0.055	−25 ± 1.32	59.06 ± 1.34
GC5	306 ± 16	0.137 ± 0.063	−30 ± 1.008	64.07 ± 1.76
GC6	291 ± 23	0.149 ± 0.041	−32 ± 1.123	73.73 ± 1.86
GC7	550 ± 11	0.172 ± 0.042	−36 ± 0.687	45.37 ± 1.11
GC8	477 ± 18	0.352 ± 0.029	−34 ± 1.009	47.32 ± 2.06
GC9	365 ± 20	0.132 ± 0.052	−33 ± 0.956	52.12 ± 1.36

**Table 3 pharmaceutics-15-00680-t003:** Kinetic analysis of the fabricated GFT-CNP formulations.

F. Code	Zero-Order		First-Order		Diffusion		Korsemeyer-Peppas	
	R^2^	K_0_	R^2^	K_1_	R^2^	K_H_	R^2^	n
GFT-Suspension	0.9616	1.4559	0.9685	0.00707	0.9903	5.56670	0.9919	0.7015
GC1	0.8040	4.5868	0.9239	0.04450	0.9284	19.6614	0.9526	0.1988
GC2	0.7779	4.6481	0.9156	0.05003	0.9086	20.1544	0.9255	0.1674
GC3	0.7676	4.8742	0.9275	0.05908	0.9046	21.3270	0.9514	0.1661
GC4	0.7884	3.3677	0.8660	0.02397	0.9166	14.5366	0.9194	0.1748
GC5	0.7746	3.5298	0.8611	0.02640	0.9087	15.3573	0.9256	0.1657
GC6	0.8138	4.2917	0.9173	0.03730	0.9327	18.2595	0.9246	0.1982
GC7	0.7793	2.5059	0.8321	0.01520	0.9075	10.8344	0.9273	0.1674
GC8	0.7520	2.6008	0.8085	0.01620	0.8945	11.4858	0.9530	0.1547
GC9	0.7683	2.8807	0.8335	0.01885	0.9028	12.5656	0.9042	0.1551

## Data Availability

The datasets generated during and/or analyzed during the current study are available from the corresponding authors on reasonable request.

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
