# Peer review of "Anti-Tumor Activity of Orally Administered Gefitinib-Loaded Nanosized Cubosomes against Colon Cancer"

_pharmaceutics, 2023, doi:10.3390/pharmaceutics15020680_

Round 1

Reviewer 1 Report

This contribution by Mahmoud M.A. Elsayed reports on the preparation and characterization of formulations containing the drug Gefitinib (GFT) in the form of the so-called cubosomal nanoparticles (GFT-CNPs). Various compositions of GFT-CNPs were obtained to evaluate drug entrapment efficiency, particle size, polydispersity index, and in vitro release kinetics. In one specific sample, analysis by FTIR, DSC and TEM was performed. Also, some in vivo studies were carried out in selected samples such as gene expression levels of TIMP-1 and MMP-7, estimation of colon biomarkers and histopathological examination of colon tissues.

Overall, the study is extensive and well-organized, covering various aspects of formulation development attempting to find an effective composition. I believe this study may become suitable for publication in Pharmaceutics after consideration of the following issues:

i) According to the authors this study aims “To improve gefitinib bioavailability and anti-cancer effects in this work, we employed cubosomal nanoparticles (CNPs) as a drug delivery vehicle…” (p.3 last paragraph). In the first case, however, the experimental design was not directed to evaluate the bioavailability of GFT (i.e., to measure the concentration of GFT in the bloodstream), and therefore, authors should acknowledge an appropriate aim for the study.

ii) In previous literature (references 27, 29 and 32 cited here) cubosomes are proposed as solubilizing agents for sparingly soluble drugs in water. The most common components for preparation of cubosomes are gyceryl monooleate (GMO) as the lipid, and Poloxamer 407 (P407) as stabilizer. These two components were used herein for the preparation of the formulations GFT-CNPs, and therefore it is assumed that cubosomal nanoparticles are formedin the nine formulations. However, I find troublesome that only in a single “optimized” case a blurry TEM micrograph was provided as evidence of cubosome formation with semi-cubical shape. In order to assess the relevance of the shape of these particles called as cubosomal formulation, it is important to compare the morphology of other “non-optimized” formulations.

iii) Given the low percentage of GFT in the formulations, its specific IR bands are more difficult to distinguish and seemingly disappear. The spectra of GMO and GFT-CNPs are quite similar, indicating that GMO is simply the main component in the formulation. Accordingly, I think that attributing the absence of GFT bands to specific interactions is an overstatement (p. 11 last part of the first paragraph).

iv) In the section 3.5.1 the entrapment efficiency is related to the hydrophobic nature of GFT. This “hydrophobic nature” would be reasonable if the molecule is always in its neutral state but given its pKa values (5.4 and 7.2) the molecule is surely ionized (as mono- and dicationic species) in the pH interval at/below 7, then, what is indeed the driving force for entrapment? Authors may acknowledge and discuss what is the role of the charge state of GFT within this lipid vesicular system.

v) In the section 3.5.3 the kinetics release profile is described as bi-phasic with an “initial burst release occurs within the first 2h followed by a sustained release pattern over 12h.” In the same paragraph an explanation is provided as “The initial release may be due to the diffusion of the un-entrapped GFT that could be easily diffused to the matrix, while the sustained release pattern may be attributed to the diffusion of GFT through the water channels within the cubic nanoparticles.” However, if the burst is due to un-entrapped GFT, it must have an inverse correlation with the total amount of GFT entrapped, and Figure 10 indicates otherwise. After the initial “burst” (which basically is a single point) the release seems to have a very similar kinetic profile, independently of the fact if formulation is optimized or un-optimized.

Please comment what would be the effect on the release kinetics experiments if the particles used were washed with saline solution to assure that un-entrapped GFT is no longer present (as described for the entrapment study in the section 2.3.3.1).

vi)  I suggest revising the use of significant figures included in the text of section 3.5.1, 3.5.2 and Table 2. As an example, reporting data such as 149.21 ± 24.11 is not correct, and it should read as 149 ± 24.

Author Response

Reviewer 1

 Dear reviewer,

We want to extend our appreciation for taking the time and effort necessary to provide such insightful guidance. The revision, based on the review team’s collective input, includes a number of positive changes. Based on your guidance, we have accordingly, the manuscript was edited as required and corrections were highlighted in yellow color modified the manuscript, and detailed corrections, changes and/or rebuttals against each point raised are listed below.

  1. i) According to the authors this study aims “To improve gefitinib bioavailability and anti-cancer effects in this work, we employed cubosomal nanoparticles (CNPs) as a drug delivery vehicle…” (p.3 last paragraph). In the first case, however, the experimental design was not directed to evaluate the bioavailability of GFT (i.e., to measure the concentration of GFT in the bloodstream), and therefore, authors should acknowledge an appropriate aim for the study.

 The response i: The aim of the study was amended to (To investigate the gefitinib anti-cancer effects) instead of (bioavailability enhancement)

  1. ii) In previous literature (references 27, 29, and 32 cited here) cubosomes are proposed as solubilizing agents for sparingly soluble drugs in water. The most common components for the preparation of cubosomes are glyceryl monooleate (GMO) as the lipid, and Poloxamer 407 (P407) as a stabilizer. These two components were used herein for the preparation of the formulations GFT-CNPs, and therefore it is assumed that cubosomal nanoparticles are formed in the nine formulations. However, I find it troublesome that only in a single “optimized” case a blurry TEM micrograph was provided as evidence of Cubosome formation with semi-cubical shape. In order to assess the relevance of the shape of these particles called as cubosomal formulation, it is important to compare the morphology of other “non-optimized” formulations.

 Response ii: TEM of plain cubosomes (A), GC8 (B), and GC9 (C) were performed for comparison of the formulated cubosomal formulae morphology.

Figure 4 was changed in the manuscript.

Figure 4. TEM micrograph of A: GFT-CNPs plain optimized formulation (GC9), B: GFT-CNPs, GC8, and C: GFT-CNPs optimized formulation (GC9)

iii) Given the low percentage of GFT in the formulations, its specific IR bands are more difficult to distinguish and seemingly disappear. The spectra of GMO and GFT-CNPs are quite similar, indicating that GMO is simply the main component in the formulation. Accordingly, I think that attributing the absence of GFT bands to specific interactions is an overstatement (p. 11 last part of the first paragraph).

Response iii: the paragraph (The FTIR spectra of the optimized GFT-CNPs formula (GC9) showed a slight shift and decreased peak intensity of the characteristic CN and NH groups of pure GFT, while the peak of the C=C group disappeared) was changed in the text to (The FTIR spectra of the optimized GFT-CNPs formula (GC9) showed a slight shift, decreased peak intensity, and disappearance of some GFT characteristic peaks).

The paragraph (This result is in agreement with their DSC thermograms and is attributed to the possible interaction between the hydroxyl group of GMO and the pure drug NH group) was changed in the text to (This result is in agreement with their DSC thermograms and is attributed to the possible interaction between GMO and GFT)

  1. iv) In the section 3.5.1 the entrapment efficiency is related to the hydrophobic nature of GFT. This “hydrophobic nature” would be reasonable if the molecule is always in its neutral state but given its pKa values (5.4 and 7.2) the molecule is surely ionized (as mono- and dicationic species) in the pH interval at/below 7, then, what is indeed the driving force for entrapment? Authors may acknowledge and discuss what is the role of the charge state of GFT within this lipid vesicular system.

Response iv: The EE% was found to be affected by the concentration of P407 beside the lipophilicity of the GFT. As the concentration of P407 increased, there observed increase in EE%. When the content of poloxamer 407 increased, the ability of the formulated CNPs to hold GFT increased possibly due to the increased hydrophilicity of the GFT-CNPs and entrapment of GFT into the aqueous core together with P407.

The paragraph (The high EE% of GFT may be attributed to its highly hydrophobic nature in addition to GMO/P407 ratio and P407 concentration) was deleted.

And the text was changed to The EE% of all GFT-CNPs formulations are represented in Figure 5. The EE% was in the range between 45.24 ± 3.87 and 83.11 ± 4.39. The main effects plot and response surface figure (Figure 6) showed that the GFT-CNPs containing a higher concentration of P407 have higher EE% than those containing lower concentrations. The EE% was found to be affected by the concentration of P407 beside the lipophilicity of the GFT. As the concentration of P407 increased, there observed an increase in EE%. When the content of poloxamer 407 increased, the ability of the formulated CNPs to hold GFT increased possibly due to the increased hydrophilicity of the GFT-CNPs and entrapment of GFT into the aqueous core together with P407. The obtained results were in agreement with many studies showing that P407 (a surfactant with low hydrophilic-lipophilic balance) is the stabilizer of choice in cubosomes preparation

  1. v) In the section 3.5.3 the kinetics release profile is described as bi-phasic with an “initial burst release occurs within the first 2 h followed by a sustained release pattern over 12 h.” In the same paragraph an explanation is provided as “The initial release may be due to the diffusion of the un-entrapped GFT that could be easily diffused to the matrix, while the sustained release pattern may be attributed to the diffusion of GFT through the water channels within the cubic nanoparticles.” However, if the burst is due to un-entrapped GFT, it must have an inverse correlation with the total amount of GFT entrapped, and Figure 10 indicates otherwise. After the initial “burst” (which basically is a single point) the release seems to have a very similar kinetic profile, independently of the fact if formulation is optimized or un-optimized.

Please comment what would be the effect on the release kinetics experiments if the particles used were washed with saline solution to assure that un-entrapped GFT is no longer present (as described for the entrapment study in the section 2.3.3.1).

Response v:

The in vitro release studies were performed utilizing the formulated CNPs batches without centrifugation so the higher initial burst release was suggested to be due to the adsorbed GFT or weakly bound GFT to the relatively larger surface of GFT-CNPs. The burst release of hydrophilic and hydrophobic drugs from cubosomes was previously reported.

Reference: Ali Z, Sharma PK, Warsi MH. Fabrication and Evaluation of Ketorolac Loaded Cubosome for Ocular Drug Delivery. J App Pharm Sci, 2016; 6 (09): 204-208. 

  1. vi)  I suggest revising the use of significant figures included in the text of section 3.5.1, 3.5.2 and Table 2. As an example, reporting data such as 149.21 ± 24.11 is not correct, and it should read as 149 ± 24.

Response vi: the data was revised.

Reviewer 2 Report

The authors Ahmed A. El-Shenawy et al. discuss the anti-tumor activity of orally administered gefitinib-loaded nanosized cubosomes against colon cancer. The manuscript needs serious corrections and modifications before going for publication in Pharmaceutics. Below are my major concerns:

a)       The authors did not show the effect of in-vitro efficacy /cell viability of cubosome formulations. It is very difficult to understand whether these cubosomes retain the anti-tumor effect of GFT.

b)      It is advised to perform an anti-tumor in vivo study with tumor measurement with the same formulations. Reduction of expression of serum level is a secondary result.

c)       Discussion for each experiment was unsatisfactory. There was an overlap with the introduction section. And, the discussion seemed to be the simple explanation for the results.  It should be provided in the manuscript at length.

d)      The authors didn’t show why the formulation GC9 was chosen over other formulations for further evaluation such as stability studies, and animal studies. There is not much difference in GC8 and GC9 formulation in in-vitro drug release study. Same in the case of EE studies.

e)      The authors failed to provide the reason for the increase in vesicle size/ reduction in the EE% at 25 C in stability studies. They should explain the reason for this.

f)        Figure 14 is difficult to interpret. Please put a scale bar and provide another figure for 14d. Figure 14e looks zoomed out.

g)       Please show the statistical difference between the DMH+GFT suspension and DMH+GFT cubosomes. If both formulations are given via oral gavage, there is a very narrow difference in the two formulations for CA19.9 and CEA serum levels. The statistical bar should be presented in figure 13 between each formulation.

h)      The authors mentioned in the introduction “To improve gefitinib bioavailability and anti-cancer effects in this work, we employed cubosomal nanoparticles (CNPs) ….”. However, it was not disclosed anywhere how the bioavailability is improved against GFT alone.

i)        The authors mentioned the line twice in the manuscript, “This study firmly suggests the higher anti-proliferative activity of GFT-CNPs as an oral vesicular system for the treatment of colon cancer.”. However, there is no experimental proof for this statement. 

Author Response

Dear reviewer,

We want to extend our appreciation for taking the time and effort necessary to provide such insightful guidance. The revision, based on the review team’s collective input, includes a number of positive changes. Based on your guidance, we have accordingly, the manuscript was edited as required and corrections were highlighted in yellow color modified the manuscript, and detailed corrections, changes and/or rebuttals against each point raised are listed below.

  1. a)       The authors did not show the effect of in-vitro efficacy /cell viability of cubosome formulations. It is very difficult to understand whether these cubosomes retain the anti-tumor effect of GFT.

Response a: We appreciate your insightful feedback. Our main concern is to optimize the in vitro release and formulation parameters, in addition we perform tumor biomarkers, histopathological studies and PCR,  But we don't have any samples to perform any more investigations. This will be considered in future study.

Many thanks for this suggestion, we will consider it in our future studies.

  1. b)      It is advised to perform an anti-tumor in vivo study with tumor measurement with the same formulations. Reduction of expression of serum level is a secondary result.

Response b: Many thanks for this suggestion,  as we mentioned we will consider it in our future studies.

  1. c)       Discussion for each experiment was unsatisfactory. There was an overlap with the introduction section. And, the discussion seemed to be the simple explanation for the results.  It should be provided in the manuscript at length.

Response c: The discussion has been modified in many sections included in :

3.3. FTIR Spectroscopy

3.4. GFT-CNPs Morphology

3.5.1. Entrapment Efficiency Percentage (EE%)

3.6. GFT-CNPs stability study

  1. d)      The authors didn’t show why the formulation GC9 was chosen over other formulations for further evaluation such as stability studies, and animal studies. There is not much difference in GC8 and GC9 formulation in in-vitro drug release study. Same in the case of EE studies.

Response d: The beast GFT-CNPs formulations were chosen according to the highest EE%, the lowest vesicle size, and cumulative % released after 12h. Depending on the obtained data F9 (composed of 7.5% w/w GMO/P407 and 15% P407) was selected as the best one and used for further investigation.

  1. e)      The authors failed to provide the reason for the increase in vesicle size/ reduction in the EE% at 25 C in stability studies. They should explain the reason for this.

Response e:

At 25 ± 2°C, GC9 cubosomal formulation could have acquired energy from the surrounding heat and light which enhance the Brownian motion of cubosomal particles. this behavior promotes the collision of cubosomal particles with each other resulting in P407 coating removal. The depletion of the P407 coating could lead to the adhesion of cubosomal particles with each other to generate deformed cubosomal particles with bigger sizes and reduced EE% [69]. Thus, storage of GC9 cubosomal formulation at refrigerated conditions is a suggested option to avoid aggregation and drug leakage during its storage.

6                                   6 9.    Nithya, R., P. Jerold and K. Siram. Cubosomes of dapsone enhanced permeation

                                    across the skin. J of Drug Del. Sci. Tech. 2018, 48, 75-781.

  1. f)        Figure 14 is difficult to interpret. Please put a scale bar and provide another figure for 14d. Figure 14e looks zoomed out.

Response f: The scale bar was added and the figure was changed.

  1. g)       Please show the statistical difference between the DMH+GFT suspension and DMH+GFT cubosomes. If both formulations are given via oral gavage, there is a very narrow difference in the two formulations for CA19.9 and CEA serum levels. The statistical bar should be presented in figure 13 between each formulation.

Response g: The statistical difference between the DMH+GFT suspension and DMH+GFT cubosomes was added and the figure was changed.

  1. h)      The authors mentioned in the introduction “To improve gefitinib bioavailability and anti-cancer effects in this work, we employed cubosomal nanoparticles (CNPs) ….”. However, it was not disclosed anywhere how the bioavailability is improved against GFT alone.

Response h: the aim of the study was amended to (To investigate the gefitinib anti-cancer effects) instead of (bioavailability enhancement)

  1. i)        The authors mentioned the line twice in the manuscript, “This study firmly suggests the higher anti-proliferative activity of GFT-CNPs as an oral vesicular system for the treatment of colon cancer.”. However, there is no experimental proof for this statement. 

Response i: this paragraph is misused and was changed in the manuscript to:

This study firmly suggests the possible usage use of GFT-CNPs as an oral vesicular system for the treatment of colon cancer.

Reviewer 3 Report

In this manuscript (pharmaceutics-2156573), the authors designed and prepared some GFT cubosomal nanoparticles (GFT-CNPs) by emulsification method. Then the synthesized GFT-CNPs size, morphology, FTIR, DSC, and in vitro release were measured. Additionally the Gene expression levels of TIMP-1 and MMP-7 by RT-PCR, estimation of colon biomarkers as serum cancer embryonic antigen (CEA) and carbohydrate antigen 19.9 (CA 19.9), and histopathological examination of colon tissues were estimated. However, the manuscript is not suitable for publishing as it current form without more detailed information. A couple of questions might help to improve this manuscript as follows:

1. The author claimed that the prepared GFT-CNPs as an oral vesicular system for the treatment of colon cancer have higher anti-proliferative activity. The prepared GFT-CNPs with higher efficiency or similar to GET. They should evaluate the in vitro antiproliferative inhibitory potency of these compounds against different human colon cancer cell lines with MTT assay. It would be more convincing than just using histopathological examination.

2. In order to more fully characterize the prepared GFT-CNPs. The authors should record the 1H and 13C NMR spectra of the prepared GFT-CNPs.

3. The reference style does not meet the requirements of Pharmaceutics, some page numbers are missing or mistaking.

Author Response

Dear reviewer,

We want to extend our appreciation for taking the time and effort necessary to provide such insightful guidance. The revision, based on the review team’s collective input, includes a number of positive changes. Based on your guidance, we have accordingly, the manuscript was edited as required and corrections were highlighted in yellow color modified the manuscript, and detailed corrections, changes and/or rebuttals against each point raised are listed below.

  1. The author claimed that the prepared GFT-CNPs as an oral vesicular system for the treatment of colon cancer have higher anti-proliferative activity. The prepared GFT-CNPs with higher efficiency or similar to GET. They should evaluate the in vitro antiproliferative inhibitory potency of these compounds against different human colon cancer cell lines with MTT assay. It would be more convincing than just using histopathological examination.

Response 1: Dear Reviewer, we thank you for your valuable comment. But we don't have samples to do that evaluation. We will take this into account in future research.

  1. In order to more fully characterize the prepared GFT-CNPs. The authors should record the 1H and 13C NMR spectra of the prepared GFT-CNPs.

Response 2: In this work, we used IR and DSC to study the interaction between the drug and the used excipients and clearly showed no interaction. In future work, we will make also 1H and 13C NMR to fully characterize our formulation.

  1. The reference style does not meet the requirements of Pharmaceutics, some page numbers are missing or mistaking.

Response 3: the reference style was changed to meet the requirements of pharmaceutics

Round 2

Reviewer 1 Report

Authors have made suitable modifications to the submitted manuscript, so I believe the current revised version is acceptable for publication.

Author Response

Authors have made suitable modifications to the submitted manuscript, so I believe the current revised version is acceptable for publication.

Reply: We want to extend our appreciation for taking the time and effort necessary to provide such insightful guidance.